# Initialize to Generalize: A Stronger Initialization Pipeline for Sparse-View 3DGS

## Abstract

Sparse-view 3D Gaussian Splatting (3DGS) often overfits to the training views, leading to artifacts like blurring in novel view rendering. Prior work addresses it either by enhancing the initialization (*i.e.*, the point cloud from Structure-from-Motion (SfM)) or by adding training-time constraints (regularization) to the 3DGS optimization. Yet our controlled ablations reveal that initialization is the decisive factor: it determines the attainable performance band in sparse-view 3DGS, while training-time constraints yield only modest within-band improvements at extra cost. Given initialization's primacy, we focus our design there. Although SfM performs poorly under sparse views due to its reliance on feature matching, it still provides reliable seed points. Thus, building on SfM, our effort aims to supplement the regions it fails to cover as comprehensively as possible. Specifically, we design: (i) frequency-aware SfM that improves low-texture coverage via low-frequency view augmentation and relaxed multi-view correspondences; (ii) 3DGS self-initialization that lifts photometric supervision into additional points, compensating SfM-sparse regions with learned Gaussian centers; and (iii) point-cloud regularization that enforces multi-view consistency and uniform spatial coverage through simple geometric/visibility priors, yielding a clean and reliable point cloud. Our experiments on LLFF and Mip-NeRF360 demonstrate consistent gains in sparse-view settings, establishing our approach as a stronger initialization strategy.

## 1 INTRODUCTION

Novel view synthesis (Mildenhall et al., 2021; Chan et al., 2023; Liu et al., 2023) plays a central role in applications such as virtual and augmented reality (Anthes et al., 2016), free-viewpoint video (Zhou et al., 2025), and digital content creation (Cao et al., 2024), where the goal is to generate realistic novel views from captured imagery. To enable such capabilities, recent advances in 3D neural scene representations (Mildenhall et al., 2021; Deng et al., 2022) have demonstrated remarkable performance, with 3D Gaussian Splatting (3DGS) (Kerbl et al., 2023) emerging as a highly efficient alternative to neural radiance fields (Mildenhall et al., 2021). In practice, however, data acquisition is often limited to a small number of viewpoints due to constraints in hardware, cost, or capture conditions. Under such sparse-view inputs, vanilla 3DGS is prone to severe overfitting to the training views (Zhu et al., 2024; Dai & Xing, 2025), resulting in noticeable artifacts and persistent floaters when rendering unseen views.

To address this challenge, existing solutions broadly fall into two categories. The first focuses on enhancing the initial point cloud of 3DGS, which is typically obtained from Structure-from-Motion (SfM) (Ullman, 1979). For instance, CoMapGS (Jang & Pérez-Pellitero, 2025) leverages a depth estimator together with a robust visual encoder to produce a stronger initialization. The second line of work constrains the growth or optimization of Gaussian primitives, often by introducing regularization losses during training. A representative example is CoR-GS (Zhang et al., 2024b), which prunes outlier primitives by enforcing consistency between two parallel Gaussian fields. While such methods have shown effectiveness in practice, our evidence suggests that training-time regularization mainly serves as a patch for poor initialization rather than addressing the fundamental bottleneck in sparse-view 3DGS.

We begin with ablation studies comparing the effectiveness of two strategies for mitigating overfitting. In Section 3, we evaluate several state-of-the-art training-time constraints across a range of

initialization strengths. These initialization strengths are explicitly controlled ideally (rather than being determined solely by the input images). The results show that initialization quality sets the attainable performance band, whereas these constraints provide only modest within-band gains. The findings indicate that initialization is the more influential lever, which motivates focusing on developing a stronger initialization pipeline.

We start from the vanilla SfM algorithm since it provides reliable initial seed points, though it produces only a few in sparse-view scenarios due to its feature-matching nature. Our work is dedicated to accurately supplementing those undercovered regions by fully exploiting the visual information contained in the available images, step by step. First, inspired by EAP-GS (Dai & Xing, 2025), we modify the standard SfM pipeline by relaxing the minimum track matching numbers from three to two, yielding a denser initial point set. Moreover, distinct from prior work, we pre-mask high-frequency regions and perform SfM on the resulting augmented, doubled image set, which encourages a more balanced and richer point distribution. Secondly, to further enrich the point set, we propose 3DGS Self-Initialization, which lifts per-pixel photometric constraints into additional points by leveraging the learning signal of 3DGS. Specifically, after training a first-pass lightweight 3DGS on the input views and reusing all primitive centers as a point cloud, we are able to compensate for regions with insufficient image features. Finally, we introduce further regularization techniques to refine the final point cloud by removing noise and unreliable points: (1) discarding points observed from only a single view due to depth ambiguity, (2) applying clustering-based noise reduction, and (3) enforcing normal-based consistency filtering.

To summarize, our contributions are as follows:

1. Our controlled ablations with explicitly set initialization strengths show that sparse-view performance in 3DGS is initialization-limited, and training-time constraints provide only modest improvement.

2. We design a rich and reliable three-stage initialization: (i) a low-frequency-aware SfM variant that relaxes the minimum track length from three to two and pre-masks high-frequency content to improve low-texture coverage; (ii) 3DGS Self-Initialization that trains a first-pass 3DGS and reuses all primitive centers as the initial point cloud, turning photometric cues into additional points; and (iii) point-cloud regularization for removing noise or unreliable points, including removal of single-view points, clustering-based denoising, and a simple geometric prior for consistency and coverage.

3. Experiments demonstrate that our proposed method achieves state-of-the-art performance on LLFF and Mip-NeRF360 datasets and offers a superior initialization choice for sparse-view 3DGS.

## 2 RELATED WORK

### 2.1 NOVEL VIEW SYNTHESIS

Novel view synthesis (NVS) aims to generate photorealistic images from novel viewpoints given only a finite set of calibrated observations of a typically static scene (Goesele et al., 2007). Among the different strategies, optimization-based methods reconstruct a full 3D scene representation, from which novel views can be rendered at arbitrary camera poses. Neural scene representations (Tewari et al., 2022) have recently emerged as the dominant paradigm: NeRF, for example, encodes view-dependent color and density and renders images through volumetric integration (Mildenhall et al., 2021). Beyond such implicit fields, explicit formulations like 3D Gaussian Splatting (3DGS) represent scenes as anisotropic Gaussians and achieve real-time rendering with competitive quality (Kerbl et al., 2023; Lin et al.; Zhang et al., 2024a). In parallel, generative approaches that exploit large-scale priors have also been developed (Liu et al., 2023; Wang et al., 2025; Tang et al., 2024; Szymanowicz et al., 2024; Voleti et al., 2024). These methods provide extremely fast inference and reduced reliance on precise camera pose annotations, but they remain limited in maintaining multi-view consistency and geometric fidelity.

**3D Reconstruction with Sparse Views.** Reconstructing 3D scenes from sparse views is inherently under-constrained: limited viewpoints lead to incomplete surface coverage, ambiguous depth estimation, and frequent artifacts such as floaters and texture misalignment. To address these challenges, some works incorporate additional geometric and photometric priors, introducing supervision such as depth (Chung et al., 2024; Zheng et al., 2025) to guide the optimization process. Another line

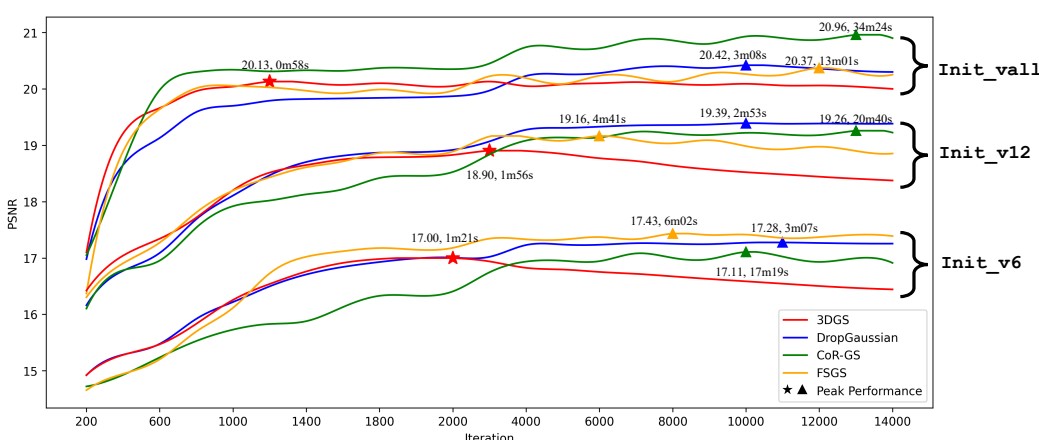

Figure 1: Ablation study on Mip-NeRF360 under the 12-view sparse-view protocol PSNR on average, comparing three training-time constraints (DropGaussian, CoR-GS, and FSGS) against vanilla 3DGS across different initialization strengths (`Init_v6`, `Init_v12`, `Init_vall`). Results show that while regularization methods modestly delay overfitting and improve performance within each initialization band, the final reconstruction quality is primarily governed by initialization strength.

leverages controllable 2D generative models to synthesize novel training views, effectively densifying the supervision and mitigating the scarcity of input observations (Mithun et al., 2025; Wan et al., 2025; Xiong et al., 2023; Chen et al., 2024b;a; Liu et al., 2024).

## 2.2 OVERFITTING IN SPARSE-VIEW 3DGS

Under sparse views, vanilla 3DGS optimized from photometric supervision tends to overfit the limited inputs, degrading markedly on unseen view rendering. Efforts to mitigate overfitting generally take two forms, the first being to enhance the initialization of the 3DGS point cloud (Bao et al., 2025; Jang & Pérez-Pellitero, 2025; Xu et al., 2025). Intuitively, initialization provides the spatial and geometric bias for subsequent optimization; with a better-distributed seed point cloud, the system is guided toward plausible geometry rather than overfitting to limited photometric cues. For instance, LoopSparseGS (Bao et al., 2025) introduces a progressive Gaussian initialization strategy that iteratively densifies the seed point cloud using pseudo-views, significantly improving coverage and stability under sparse inputs. EAP-GS (Dai & Xing, 2025) augments the SfM point cloud by relaxing track constraints and adaptively densifying underrepresented regions, providing a stronger initialization for few-shot 3DGS.

The second form constrains the optimization or growth of Gaussian primitives during training, typically by introducing regularization losses or pruning strategies to suppress overfitting artifacts (Park et al., 2025; Paliwal et al., 2024; Li et al., 2024; Xu et al., 2025). For instance, CoR-GS (Zhang et al., 2024b) introduces a co-regularization framework that enforces consistency between two parallel Gaussian fields, effectively pruning outliers and improving generalization under sparse views. FSGS (Zhu et al., 2024) imposes frequency-based structural regularization, suppressing high-frequency artifacts and stabilizing training when input views are highly limited. DropGaussian (Park et al., 2025) proposes a dropout-style scheme that randomly removes Gaussians during training, enhancing gradient flow to the remaining primitives and mitigating overfitting without extra priors.

## 3 INITIALIZATION VS. REGULARIZATION: AN EMPIRICAL STUDY

### 3.1 PRELIMINARY AND DEFINITION

**Structure-from-Motion (SfM)** (Ullman, 1979). Given images $\{I_i\}_{i=1}^{N}$ with intrinsics $\{K_i\}$, SfM estimates camera poses $P_i = (R_i, t_i)$ and a sparse 3D point set $\mathcal{X} = \{X_m \in \mathbb{R}^3\}$ from multi-

view feature tracks. Tracks link projections of the same 3D point across views. In standard SfM system (*i.e.*, COLMAP (Schonberger & Frahm, 2016)), tracks observed in at least three images are retained for triangulation and bundle adjustment, yielding a conservative but reliable reconstruction. Candidate points are obtained via two-view triangulation $T$:

$$X_m = T\big(\tilde{x}_{i,m}, \tilde{x}_{j,m}, P_i, P_j\big), \quad i \neq j, \tag{1}$$

Formally,

$$\{\hat{P}_i\}, \{\hat{X}_m\} = \arg\min_{\{P_i\}, \{X_m\}} \sum_{m: |\mathcal{V}_m| \geq 3} \sum_{i \in \mathcal{V}_m} \rho\Big(\big\|\pi(K_i, P_i, X_m) - \tilde{x}_{i,m}\big\|_2^2\Big), \tag{2}$$

where $\mathcal{V}_m = \{\, i : \text{point } m \text{ observed in image } i \,\}$, $\tilde{x}_{i,m} \in \mathbb{R}^2$ is the observed pixel, $\pi(\cdot)$ denotes the standard perspective projection with intrinsics $K_i$ and pose $(R_i, t_i)$, and $\rho(\cdot)$ is a robust loss.

**3D Gaussian Splatting (3DGS)** (Kerbl et al., 2023). 3DGS represents a scene with anisotropic 3D Gaussian primitives $\mathcal{G} = \{g_n\}_{n=1}^M$, each $g_n = (\mu_n, \Sigma_n, \alpha_n, \mathbf{c}_n)$ (mean, covariance, opacity, color). In standard practice, primitives are initialized from the SfM point cloud $\{X_m\}$, with both their positions and colors initialized from the reconstructed points, and then optimized end-to-end with fixed camera poses. During training, the primitive set is adaptively densified by splitting or cloning Gaussians to grow $\mathcal{G}$ in regions indicated by image evidence. Rendering produces predictions $\hat{I}_i$ via differentiable alpha compositing of depth-sorted 2D projections of the Gaussians, and the parameters are learned via a compact photometric objective with a structural-similarity term and regularization:

$$\min_{\mathcal{G}} \sum_{i=1}^N \Big[(1-\lambda)\sum_{p \in \Omega_i}\big\|\hat{I}_i(p) - I_i(p)\big\|_1 + \lambda\, \text{D-SSIM}(\hat{I}_i, I_i)\Big] + \beta\, \mathcal{R}_{\text{reg}}(\mathcal{G}), \tag{3}$$

where $\Omega_i$ is the pixel domain, $\lambda \in [0, 1]$ balances the terms, $I_i$ and $\hat{I}_i$ are the $i$-th ground-truth and rendered images, $\text{D-SSIM}(\hat{I}_i, I_i) = 1 - \text{SSIM}(\hat{I}_i, I_i)$, and $\mathcal{R}_{\text{reg}}$ collects mild priors on scale/opacity.

**Initialization Strength.** Ideally, a strong initialization should approximate a point cloud in which the point set fully covers all object visible surfaces. Such an ideal seed could, in principle, be obtained by running SfM on a sufficiently large set of views of the scene. Since unlimited viewpoints are not available, we approximate initialization strength per scene by varying the number of input views for SfM.

Formally, let $\mathcal{V}_{\text{all}}$ denote all available views of a scene, and let the SfM reconstruction from $\mathcal{V}_{\text{all}}$ serve as a best-available proxy for an ideal initialization, producing a point cloud $\mathcal{X}_{\text{all}}$. For a view budget $n$, we uniformly sample a subset $\mathcal{V}_n \subset \mathcal{V}_{\text{all}}$ with $|\mathcal{V}_n| = n$ in camera pose space, run SfM on $\mathcal{V}_n$, and obtain a seed point cloud $\mathcal{X}_n$. Larger $n$ therefore corresponds to higher initialization strength in expectation. On the Mip-NeRF360 (Barron et al., 2022) dataset, we instantiate three levels of initialization strength: `Init_v6`, `Init_v12`, and `Init_vall`, where the subscript denotes the number of input views used by SfM (with `Init_vall` using all available views).

## 3.2 Empirical Analysis and Findings

To investigate the causes of sparse-view overfitting, we study two common strategies: (i) improving the SfM initialization cloud and (ii) adding training-time regularization of Gaussian primitives. These strategies are treated as two factors in a controlled ablation, enabling us to isolate their individual and combined effects. Specifically, for training-time constraints, we adopt representative off-the-shelf methods to reflect current practice: FSGS (Zhu et al., 2024), Cor-GS (Zhang et al., 2024b), and DropGaussian (Zhang et al., 2024b). For initialization, rather than comparing heterogeneous approaches, we treat our designed *Initialization Strength* as a controlled variable to isolate its effect. Experiments are conducted on the Mip-NeRF360 dataset, following the standard 12-view sparse-view reconstruction protocol and reporting average PSNR across all scenes.

The performance curves are shown in Figure 1. Across all three initialization levels, different regularization methods show some effectiveness: compared with vanilla 3DGS, they mitigate overfitting by delaying the performance peak and slightly improving reconstruction quality, albeit at the cost of increased computation. Strikingly, the strength of initialization proves decisive for the final outcome. The curves stratify into distinct bands according to initialization level, within which regularization offers only limited gains. This initialization-dominated phenomenon motivates us to design a more effective initialization strategy for sparse-view 3DGS.

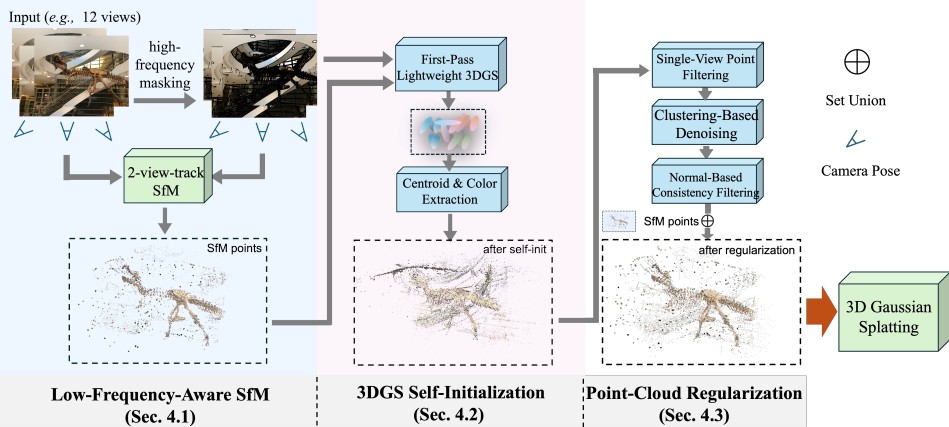

Figure 2: Our initialization pipeline for sparse-view 3DGS. Given sparse multi-view images with camera poses, we first mask high-frequency regions and perform SfM to obtain a raw point cloud, which better captures smooth areas. To compensate for regions lacking distinctive features, we train a lightweight first-pass 3DGS and incorporate its primitive centroids into the initialization. Finally, we apply clustering-based denoising, single-view filtering, and physics-based regularization to rectify the point cloud and suppress noise.

## 4   METHOD

In this section, we aim to develop a rich and reliable initialization pipeline tailored for 3D Gaussian Splatting (3DGS) under sparse-view settings. Building upon vanilla 3DGS, our objective is to compensate for undercovered regions by fully leveraging the visual information available in the input images. The overall framework is illustrated in Figure 2. It comprises three successive components: 1) Low-Frequency-Aware SfM (Sec. 4.1), which augments vanilla SfM with high-frequency masking to improve initialization in smooth regions; 2) 3DGS Self-Initialization (Sec. 4.2), which exploits the pixel-level photometric supervision of 3DGS to supplement points in regions with few distinctive features; and 3) Point-Cloud Regularization (Sec. 4.3), which applies clustering, filtering, and geometry-inspired constraints to refine the point cloud and suppress noise.

### 4.1   LOW-FREQUENCY-AWARE STRUCTURE-FROM-MOTION (SFM)

Under sparse-view capture, limited overlap makes tracks with $|\mathcal{V}_m| \geq 3$ scarce in vanilla SfM, leading to under-covered and uneven point distributions. To better exploit limited views, EAP-GS (Dai & Xing, 2025) retains two-view tracks in addition to the standard three-view requirement with fixed camera poses $\hat{P}_i$.

$$\{\hat{X}'_m\} = \underset{\{X'_m\}}{\arg\min} \sum_{m:\,|\mathcal{V}_m|\geq 2} \sum_{i\in\mathcal{V}_m} \rho\Big(\big\|\pi(K_i, \hat{P}_i, X'_m) - \tilde{x}_{i,m}\big\|_2^2\Big). \qquad (4)$$

where camera poses $\hat{P}_i$ can be either provided or estimated from SfM. Unlike EAP-GS, which identifies low-density regions and performs SfM twice merely to enlarge the point set, we adopt a more effective strategy. High-frequency regions naturally produce more track matches in SfM due to their richer image features, leading to an undesired concentration of points. To mitigate this, we pre-mask these regions and feed two image sets into SfM simultaneously, thereby generating a more balanced point cloud.

Formally, given $N$ scene views $\{I_i\}_{i=1}^N$ with camera poses $\{P_i\}_{i=1}^N$, we derive low-frequency images $\{I_{mask}\}$ by Gradient-based masking (Sobel et al., 1968). We then build an augmented view set $\mathcal{I}^{\text{aug}} = \{I_i, I_{mask}\}_{i=1}^N$, extract features on all $2N$ views, and obtain the initial 3D point set $\mathcal{P}_0$ by running SfM once with Equation 4.

## 4.2 3D GAUSSIAN SPLATTING SELF-INITIALIZATION

While SfM reconstructs 3D points by detecting and matching keypoints across views, its reliance on local features causes failures in weakly textured or repetitive regions. To overcome this bottleneck, we introduce a novel 3DGS self-initialization method, which elevates pixel-level photometric supervision into 3D space and constructs complementary point clouds by repurposing primitive centers as new points.

**Light-weight 3DGS.** We train a light-weight first-pass 3DGS $\mathcal{G}^{(0)}$ on downsampled images seeded by $\mathcal{P}_0$, aiming to convert dense photometric cues into additional 3D points rather than high-fidelity rendering. We use an economical parameterization and schedule: SH degree 0 (DC color) with a short optimization on downsampled low-resolution inputs. Training stops when densification plateaus or a small step budget is reached. We then form a colored point set by taking, for each Gaussian primitive $g_n$, the primitive center and its DC color, $i.e.$, $\mathcal{P}_1 = \{(X_n, C_n)\}_{n=1}^M$ with $X_n = \mu_n$ and $C_n = \mathbf{c}_n$.

## 4.3 POINT CLOUD REGULARIZATION

Before feeding the merged points into the final 3DGS optimization, we regularize the initial colored point cloud $\mathcal{P}_{\text{init}} = \{(X_k, C_k)\}$ obtained by combining SfM points and light-weight 3DGS points, $\mathcal{P}_{\text{init}} = \mathcal{P}_0 \cup \mathcal{P}_1$. While this union improves coverage, it also aggregates errors from both sources: (i) 3DGS-generated points with only single-view supervision, which lack geometric consistency. (ii) noisy/duplicated points introduced by 3DGS split/clone densification; and (iii) outliers from unstable two-view tracks; To obtain a reliable and uniformly distributed point set, we introduce three complementary procedures: *single-view point filtering*, *clustering-based denoising*, and *normal-based consistency filtering*. These procedures operate on disjoint criteria, so the order of application has a negligible effect on the point set. We present the pseudocode in the Appendix Sec A.3.

**Single-view Point Filtering.** During 3DGS self-initialization, points supervised by a single view suffer from inherent depth ambiguity and are thus relatively unreliable. Nevertheless, their reliability is not uniform: among single-view-supported points, those closer to regions with two-view support are more trustworthy, as they are typically produced by densifying (splitting or cloning) points already supported by multiple views. In other words, proximity to accurate two-view-supported points mitigates depth ambiguity. To balance preserving single-view information with reducing noise, we retain only the top 20% of single-view-supported points with the highest reliability. To formalize this process simply:

Starting from the colored point cloud $\mathcal{P}_{\text{init}} = \{X_k, C_k\}$, We split it into single-view and multi-view-supported subsets by camera projection:

$$\mathcal{P}_{\text{init}} = \mathcal{P}_{\text{sv}} \cup \mathcal{P}_{\text{mv}}. \tag{5}$$

For each single-view point $X \in \mathcal{P}_{\text{sv}}$, we assign a reliability score $r(X)$ based on its proximity to $\mathcal{P}_{\text{mv}}$. Finally, we retain only the top fraction (i.e., 20%) of $\mathcal{P}_{\text{sv}}$ with the highest $r(X)$:

$$\mathcal{P}_{\text{sv}}^* = \text{Top}_{20\%}\{ X \in \mathcal{P}_{\text{sv}} \mid r(X) \}. \tag{6}$$

The resulting point cloud is then

$$\mathcal{P}_{\text{filter}} = \mathcal{P}_{\text{mv}} \cup \mathcal{P}_{\text{sv}}^*. \tag{7}$$

**Clustering-Based Point-Set Denoising.** To reduce the noisy points introduced from unstable 2-view SfM tracks and duplicated points from densify process in 3DGS self-initialization, we propose a denoising technique based on the clustering algorithm, which discards 70% of the points. To formalize:

Given the single-view filtered cloud $\mathcal{P}_{\text{filter}} = \{X_k, C_k\}$, we apply $K$-means clustering with $K = 1000$ clusters. For each cluster $c$ with centroid $\mu_c$ and size $|\mathcal{Q}_c|$, we retain 30% nearest points to $\mu_c$. The cluster-filtered cloud is

$$\mathcal{P}_{\text{clu}} = \big\{ X_k \in \mathcal{P}_{\text{filter}} \,; k \in \text{Top}_{30\%}\{ \|X_k - \mu_c\|_2 : X_k \in \mathcal{Q}_c\}, c = 1, \dots, K \big\}. \tag{8}$$

**Normal-based Consistency Filtering.** Finally, we remove geometrically inconsistent points by enforcing local normal agreement in 3D space. Starting from the cluster-filtered cloud $\mathcal{P}_{\text{clu}} = $

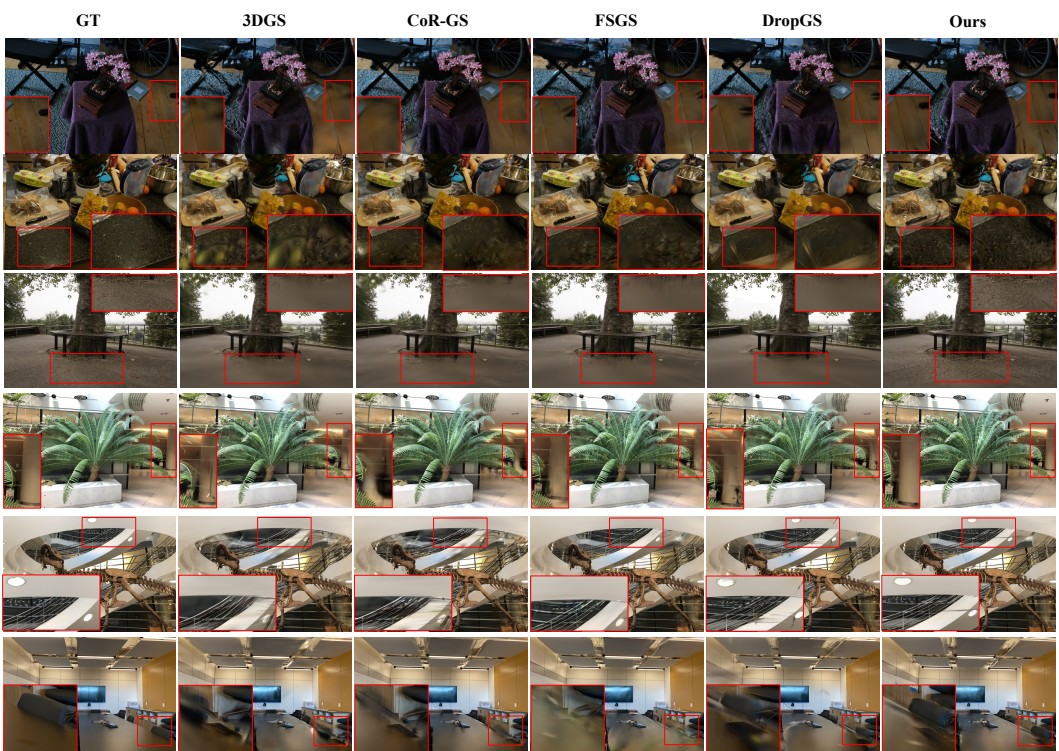

Figure 3: Qualitative comparisons on Mip-NeRF360 dataset and LLFF dataset.

$\{(X_k, C_k)\}$, we estimate a normal $n_k$ for each point via PCA on its 10 nearest neighbors in Euclidean space. Let $\mathcal{N}_k$ denote the neighbor set and assume all normals are unit-length. We compute the mean cosine similarity

$$\bar{c}_k \;=\; \frac{1}{|\mathcal{N}_k|} \sum_{j \in \mathcal{N}_k} n_k^\top n_j, \tag{9}$$

and retain a point if $\bar{c}_k \geq 0.2$. This rejects isolated outliers and unstable estimates whose normals disagree with the local surface. The final set is

$$\mathcal{P}_{\text{final}} \;=\; \{(X_k, C_k) \in \mathcal{P}_{\text{clu}} \;\mid\; \bar{c}_k \geq 0.2\}, \tag{10}$$

which yields a clean, geometrically consistent point cloud.

## 5 EXPERIMENTS

### 5.1 EXPERIMENTAL SETTINGS

**Evaluation Datasets.** We evaluate our method on two representative benchmarks, Mip-NeRF360 (Barron et al., 2022) and LLFF (Mildenhall et al., 2019). Following the conventional sparse-view 3DGS setup (Zhu et al., 2024), we use 12 uniformly distributed input views for Mip-NeRF360 and 3 views for LLFF. The camera poses are provided by the datasets. For evaluation, every eighth image is reserved as test views, and input images are downsampled by a factor of $4\times$ on both datasets.

**Metrics.** For quantitative comparison, we report three widely-used metrics: peak signal-to-noise ratio (PSNR), structural similarity index (SSIM) (Wang et al., 2004), and learned perceptual image patch similarity (LPIPS) (Zhang et al., 2018).

Table 1: Comparison on Mip-NeRF360 and LLFF datasets. We color each cell as best , second best , and third best . †: We reimplemented vanilla 3DGS with Multi-view Stereo (MVS, a native dense reconstruction stage in COLMAP following SfM) initialization, which is also deployed in FSGS, CoR-GS, and DropGS in their official implementations. It brings sizeable performance gains but increases the initialization cost. Reported time includes both initialization and training, averaged over all datasets.

| Methods | Mip-NeRF360 Dataset | | | | | LLFF Dataset | | | | |
|---|---|---|---|---|---|---|---|---|---|---|
| | PSNR | SSIM | LPIPS | Time | Iter | PSNR | SSIM | LPIPS | Time | Iter |
| 3DGS† | 19.24 | 0.5743 | 0.3660 | 8m56s | 5k | 18.95 | 0.6464 | 0.1862 | 4m00s | 5k |
| FSGS | 19.25 | 0.5719 | 0.4072 | 12m42s | 10k | 19.88 | 0.6120 | 0.3400 | 19m35s | 10k |
| CoR-GS | 19.52 | 0.5580 | 0.4180 | 39m11s | 30k | 19.45 | 0.6520 | 0.2664 | 19m45s | 10k |
| EAP-GS | 19.21 | 0.5721 | 0.3072 | 4m27s | 5k | 18.84 | 0.6358 | 0.1768 | 5m38s | 5k |
| DropGS | 19.74 | 0.5770 | 0.3640 | 11m25s | 10k | 19.54 | 0.6549 | 0.1856 | 7m11s | 10k |
| Ours | 19.77 | 0.5892 | 0.3374 | 10m48s | 5k | 19.60 | 0.6681 | 0.1852 | 5m46s | 5k |
| +DropGS | 20.07 | 0.5992 | 0.3276 | 10m38s | 10k | 19.91 | 0.6835 | 0.1659 | 8m33s | 10k |

Table 2: Ablation study on the Mip-NeRF360 dataset. Each module is incrementally added to vanilla 3DGS.

| Configuration | PSNR↑ | SSIM↑ | LPIPS↓ |
|---|---|---|---|
| **vanilla 3DGS** | 18.52 | 0.5230 | 0.4150 |
| + Low-Frequency-Aware SfM (Sec. 4.1) | $19.25^{\uparrow 0.73}$ | $0.5758^{\uparrow 0.053}$ | $0.3575^{\downarrow 0.575}$ |
| + 3DGS Self-Initialization (Sec. 4.2) | $19.42^{\uparrow 0.17}$ | $0.5924^{\uparrow 0.017}$ | $0.3246^{\downarrow 0.329}$ |
| + Single-View Point Filtering (Sec. 4.3) | $19.49^{\uparrow 0.07}$ | $0.5872^{\downarrow 0.005}$ | $0.3363^{\uparrow 0.011}$ |
| + Clustering-Based Denoising (Sec. 4.3) | $19.61^{\uparrow 0.12}$ | $0.5910^{\downarrow 0.004}$ | $0.3356^{\downarrow 0.001}$ |
| + Normal-Based Consistency Filtering (Sec. 4.3) | $19.77^{\uparrow 0.16}$ | $0.5892^{\downarrow 0.002}$ | $0.3374^{\uparrow 0.002}$ |

**Baselines.** To demonstrate the effectiveness of our initialization strategy, we compare against several state-of-the-art 3DGS-based methods, including FSGS (Zhu et al., 2024), CoR-GS (Zhang et al., 2024b), EAP-GS (Dai & Xing, 2025), and DropGaussian (Park et al., 2025).

**Implementation Details.** Following the official training set-ups of each baseline, we train FSGS, CoR-GS, EAP-GS, and DropGaussian for 10k, 10k, 5k, and 10k iterations, respectively, on the Mip-NeRF360 dataset, and for 10k, 30k, 5k, and 10k iterations on the LLFF dataset. For vanilla 3DGS and our method, we adopt 5k iterations on both benchmarks. COLMAP (Schonberger & Frahm, 2016) is configured with the same parameters as FSGS (Zhu et al., 2024) to initialize all baselines, except for EAP-GS and our approach, which are initialization-oriented methods. **For each baseline, we report the better result between our reimplementation and the original reported performance**, to ensure a fair and representative comparison. All experiments are conducted on the same hardware with a single NVIDIA RTX 4090 GPU.

## 5.2 PERFORMANCE EVALUATION.

**Qualitative Results.** We report quantitative results on the Mip-NeRF360 and LLFF datasets in Table 1. The experiments demonstrate that our initialization method alone already achieves state-of-the-art overall performance across key metrics and two datasets. Moreover, when combined with DropGS, our approach achieves further performance gains, indicating that the proposed initialization can effectively raise the upper bound of sparse-view 3DGS. We report the total time, including both initialization and training. Our method incurs a comparable time cost to existing baselines. The time cost breakdown can be found in Appendix Sec A.2.

| GT | SfM | EAP-GS | Ours |
|----|-----|--------|------|

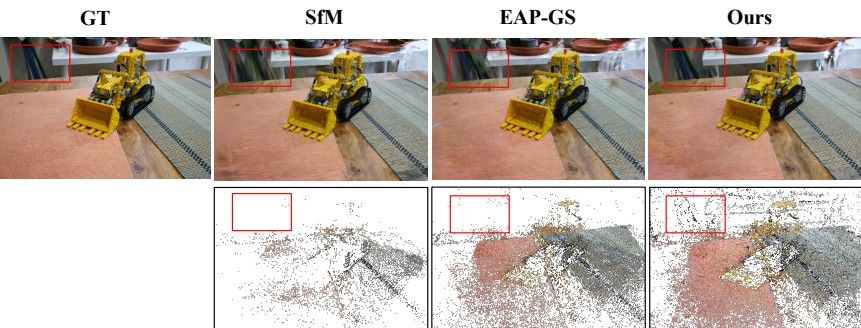

Figure 4: Comparison of initialization quality and its impact on reconstruction performance. Our method provides more accurate initialization points, especially around scene edges, leading to improved final renderings.

We further compare the performance curves of our method against baseline methods on the Mip-NeRF360 dataset, as shown in Figure 5. It demonstrates that our initialization not only achieves superior final performance but also enables faster convergence.

**Qualitative Results.** The qualitative comparisons are presented in Figure 3. Our method demonstrates consistently superior performance: it reconstructs more balanced textures in low-frequency regions (row 1), achieves robust recovery in less-featured areas (rows 2 and 3), and produces sharper object boundaries (rows 3-5) by converting pixel-level variations into reliable seed points through 3DGS self-initialization. More visualization can be found in the Appendix Sec A.2.

**Initialization v.s. Performance.** Figure 4 illustrates the relationship between final reconstruction quality and three initialization strategies: vanilla SfM, EAP-GS, and ours. Our method generates more robust initial points at scene boundaries, leading to consistently superior performance. More visualization can be found in the Appendix Sec A.2.

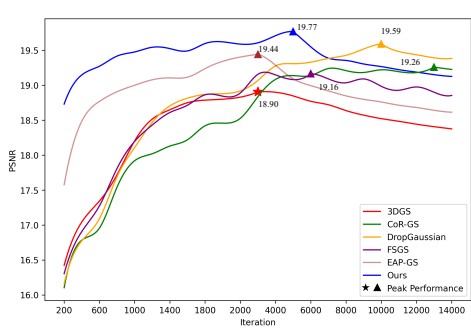

Figure 5: Performance curves between ours and baselines on the Mip-NeRF360 dataset.

### 5.3 ABLATION STUDY

We conduct an ablation study to evaluate the contribution of each component in our initialization pipeline to the final performance metrics, as shown in Table 2. The proposed Low-Frequency-Aware SfM (Sec. 4.1) and 3DGS Self-Initialization (Sec. 4.2) yield the largest improvements across all metrics, as they significantly increase the number of reliable initial points, particularly in low-texture and feature-sparse regions. Furthermore, our point cloud regularization techniques (Sec. 4.3) effectively suppress noise and redundancy in the initialization, leading to higher PSNR.

## 6 CONCLUSION

We show that sparse-view 3DGS is fundamentally initialization-limited: while training-time regularization offers only modest gains, the quality of the seed point cloud determines the achievable performance. To address this, we propose a three-stage initialization pipeline, low-frequency-aware SfM, 3DGS self-initialization, and point-cloud regularization that yields cleaner, denser, and more reliable points. Experiments on LLFF and Mip-NeRF360 confirm that our method not only surpasses prior approaches but also synergizes with existing regularization techniques, setting a stronger foundation for sparse-view novel view synthesis.

**Ethics Statement:** This work aims to advance the development of sparse-view 3D Gaussian reconstruction. While the method has positive academic value, potential risks include misuse in sensitive scenarios, the inheritance and amplification of dataset biases, and environmental impact due to computational cost. To mitigate these risks, we rely only on publicly available datasets, clearly document data sources and preprocessing steps, restrict the method to academic research purposes, and adopt efficiency-oriented experimental settings to reduce energy consumption.

**Reproducibility Statement:** We have taken concrete measures to ensure reproducibility: all datasets are publicly available, preprocessing steps are described in the main text, and all hyperparameter settings are reported in the main text.

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

# A APPENDIX

## A.1 THE USE OF LARGE LANGUAGE MODELS

We used large language models to assist in code refinement and in polishing the writing of this manuscript.

## A.2 ADDITIONAL COMPARISONS

**Performance Visualization.** We present additional visualization comparisons on Mip-NeRF360 dataset and LLFF datasets in Figure 6 and Figure 7, respectively.

**Initialization Performance Visualization.** We present additional initialization performance comparisons on Mip-NeRF360 dataset and LLFF datasets in Figure 8 and Figure 9, respectively.

**Time Cost Breakdown.** The detailed time cost between our method and baselines is presented in Table 3.

Table 3: Time Cost Breakdown.

| Methods | Mip-NeRF360 Dataset | | | LLFF Dataset | | |
|---------|------|----------|-------|------|----------|-------|
| | Init. | Training | Total | Init. | Training | Total |
| 3D-GS | 5m04s | 3m52s | 8m56s | 1m32s | 2m28s | 4m00s |
| FSGS | 5m04s | 7m38s | 12m42s | 1m32s | 18m03s | 19m35s |
| CoR-GS | 5m04s | 34m07s | 39m11s | 1m32s | 18m13s | 19m45s |
| EAP-GS | 2m15s | 4m27s | 6m42s | 2m23s | 3m15s | 5m38s |
| DropGS | 5m04s | 6m21s | 11m25s | 1m32s | 5m39s | 7m11s |
| Ours | 6m06s | 4m42s | 10m48s | 4m16s | 1m30s | 5m46s |
| Ours+DropGS | 6m06s | 4m32s | 10m38s | 4m16s | 4m17s | 8m33s |

## A.3 THE PSEUDOCODE OF SEC. 4.3.

---
**Algorithm 1** Point Cloud Regularization

---
1: **Input:** Initial colored point cloud $\mathcal{P}_{\text{init}} = \mathcal{P}_0 \cup \mathcal{P}_1$
2: **Output:** Regularized point cloud $\mathcal{P}_{\text{final}}$
3: **Step 1: Single-view Point Filtering:**
4: Split $\mathcal{P}_{\text{init}}$ into $\mathcal{P}_{\text{sv}}$ (single-view) and $\mathcal{P}_{\text{mv}}$ (multi-view)
5: **for** each $X \in \mathcal{P}_{\text{sv}}$ **do**
6:     Compute reliability score $r(X)$ w.r.t. proximity to $\mathcal{P}_{\text{mv}}$
7: **end for**
8: Retain top-20% of $\mathcal{P}_{\text{sv}}$ ranked by $r(X)$, denoted $\mathcal{P}_{\text{sv}}^*$
9: $\mathcal{P}_{\text{filter}} \leftarrow \mathcal{P}_{\text{mv}} \cup \mathcal{P}_{\text{sv}}^*$
10: **Step 2: Clustering-based Denoising:**
11: Apply $K$-means clustering ($K = 1000$) on $\mathcal{P}_{\text{filter}}$
12: **for** each cluster $c$ with centroid $\mu_c$ and points $\mathcal{Q}_c$ **do**
13:     Retain nearest-30% points to $\mu_c$
14: **end for**
15: $\mathcal{P}_{\text{clu}} \leftarrow \bigcup_c$ (retained points from $\mathcal{Q}_c$)
16: **Step 3: Normal-based Consistency Filtering:**
17: **for** each $X \in \mathcal{P}_{\text{clu}}$ **do**
18:     Estimate surface normal $n(X)$
19:     **if** angular deviation from neighbors $> \tau$ **then**
20:         Discard $X$
21:     **end if**
22: **end for**
23: $\mathcal{P}_{\text{final}} \leftarrow$ remaining points

---

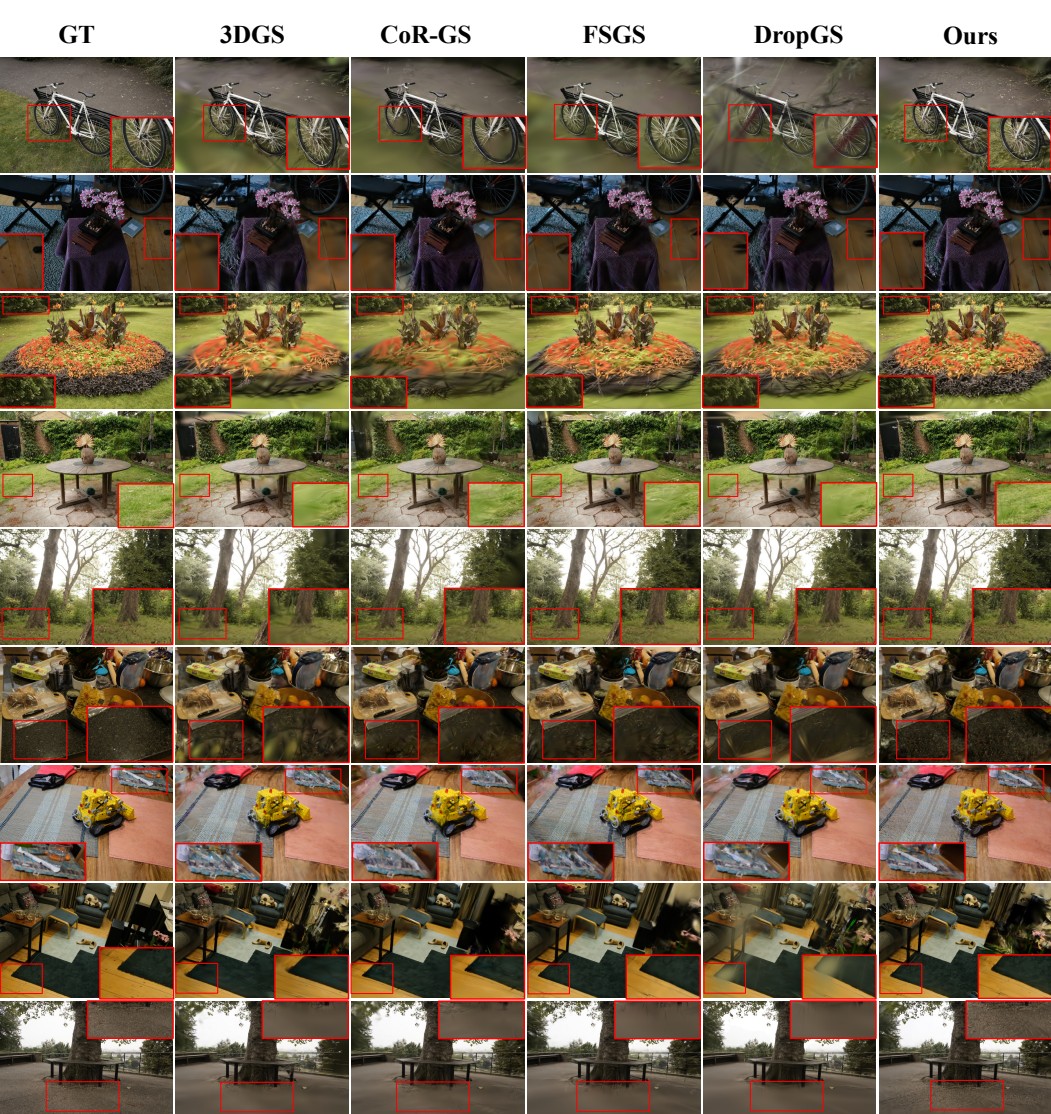

GT    3DGS    CoR-GS    FSGS    DropGS    Ours

Figure 6: Qualitative comparisons on Mip-NeRF360 dataset.

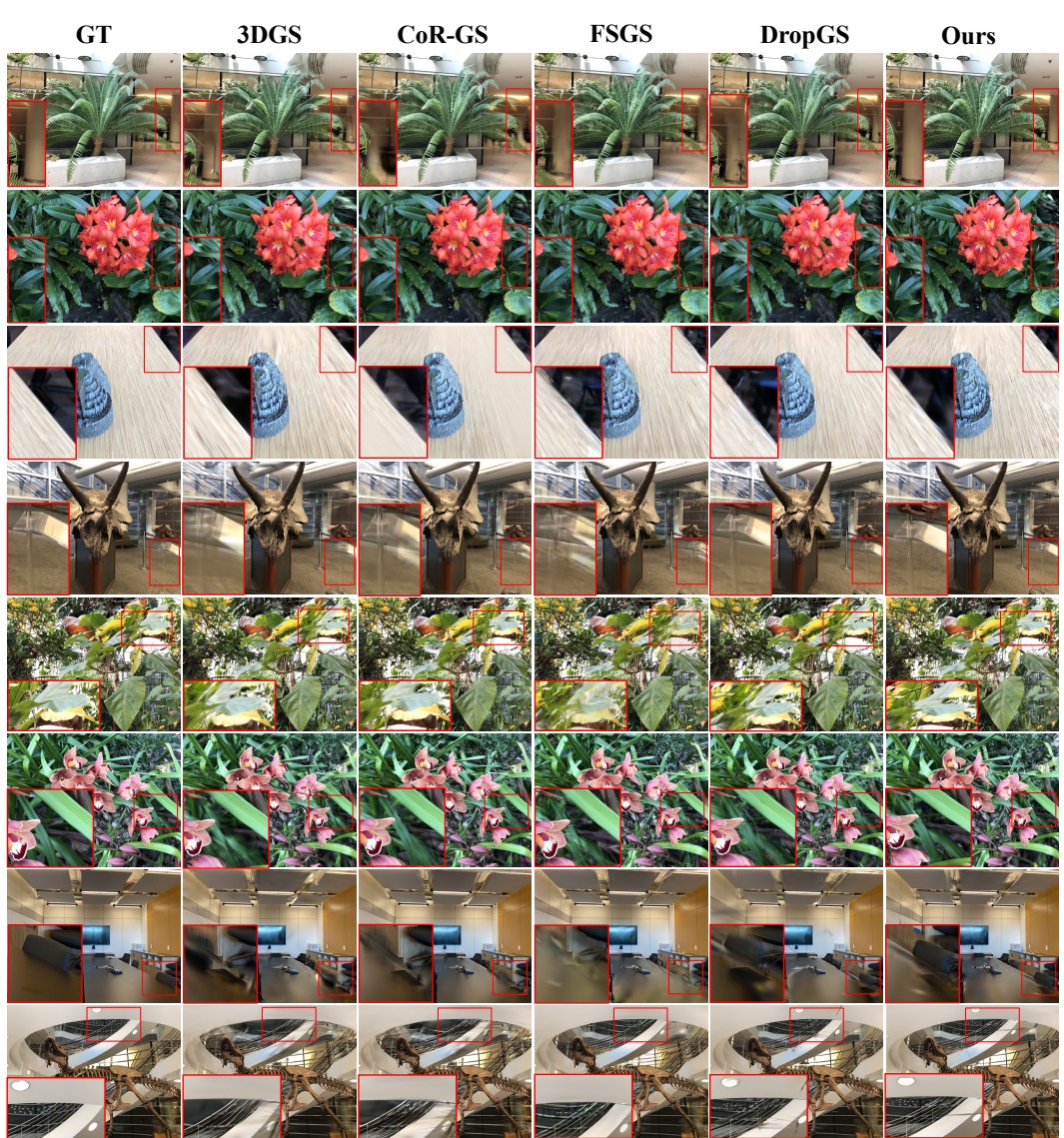

Figure 7: Qualitative comparisons on LLFF dataset.

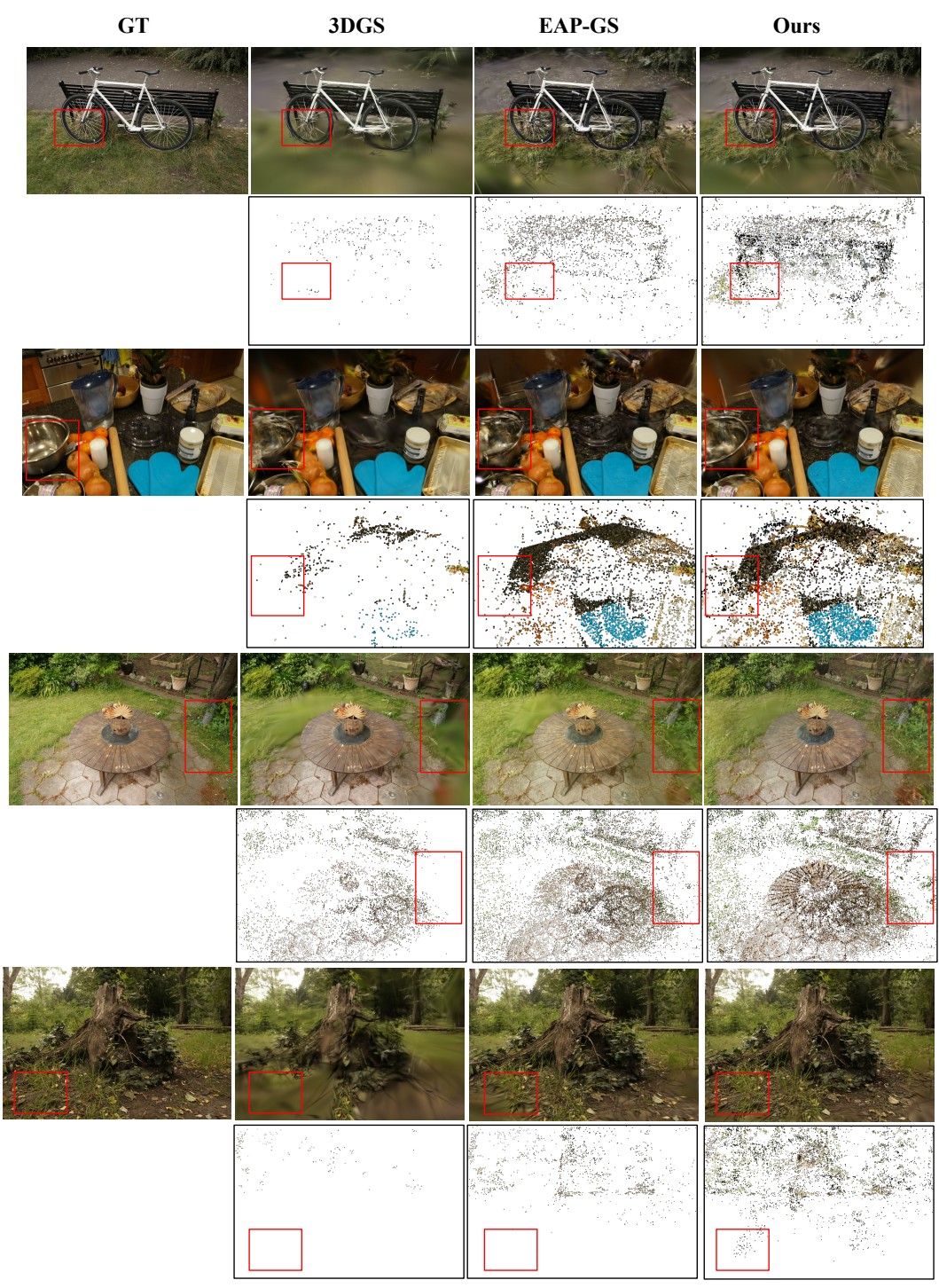

Figure 8: Initailization comparisons on Mip-NeRF360 dataset.

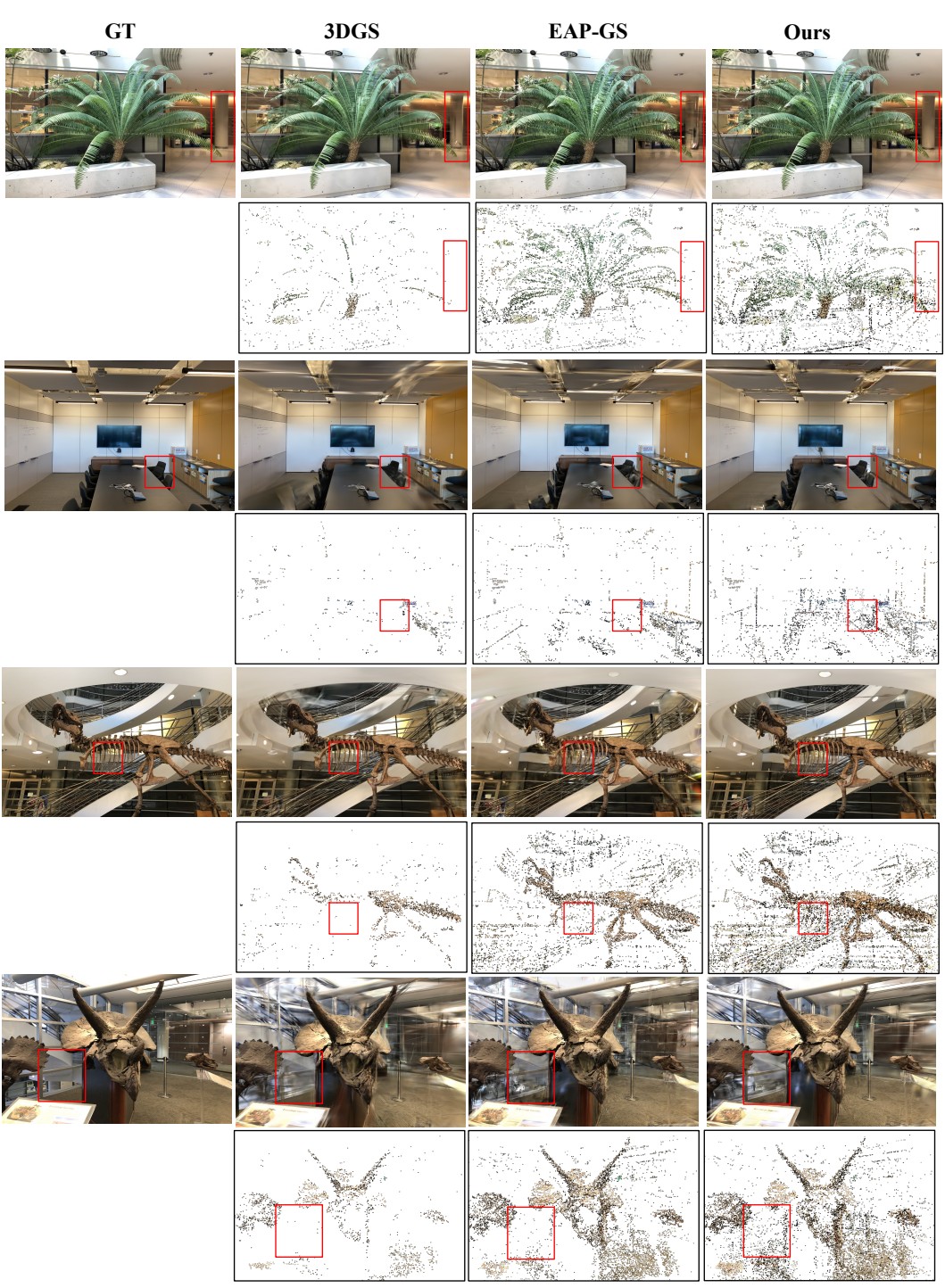

Figure 9: Initialization comparisons on Mip-NeRF360 dataset.

