# OpenReview forum: "Initialize to Generalize: A Stronger Initialization Pipeline for Sparse-View 3DGS"
_ICLR.cc/2026/Conference — ICLR 2026 Conference Withdrawn Submission_

### Official Review · Reviewer_ZJao · 2025-10-28

**Soundness:** 3
**Presentation:** 3
**Contribution:** 2
**Rating:** 4
**Confidence:** 3

**Summary:**

This work focuses on the sparse-view 3DGS task. This work points out that the initialized point cloud plays a more decisive role in the final reconstruction quality. So, this work proposes the frequency-aware SfM for improving low-texture coverage, the 3DGS self-initialization for expanding photometric supervision, and the regularization of multi-view consistency. Experimental results on different datasets show the improvements of the proposed method compared to different baselines.

**Strengths:**

* This work is well-written. The introduction of the point for "initialized point cloud" is good, as shown in Figure 1.
* The experiments are well-conducted, which provide sufficient experimental results for the claims this work proposes.
* The proposed frequency-aware SfM is interesting for me.

**Weaknesses:**

* This work mainly proposes an interesting engineering technique for initializing point clouds, but the rendering results shown in Figures 6 and 7 do not seem to have a significant gap in reconstruction quality compared to existing methods.
* This work claims that all hyper-parameter settings are reported in the main text. But, I only saw the hyperparameter settings for iterations in "Implementation Details". More hyper-parameter settings should be reported.
* More ablations of hyper-parameters are required.

**Questions:**

* I'm curious why frequency-aware SfM can achieve better reconstruction results. Could you provide a more detailed explanation?
* Following up on the previous question, my other question is that low-frequency-aware SfM masks the high-frequency part, but my understanding is that SfM requires some feature points with high-frequency information for calculation. If we mask out the high-frequency foreground of a low-frequency background, is this reasonable?
* I'm curious whether the method proposed in this work can be combined with external priors (such as monocular depth) to obtain better reconstruction results.

---

### Official Review · Reviewer_M1bt · 2025-10-31

**Soundness:** 2
**Presentation:** 3
**Contribution:** 2
**Rating:** 2
**Confidence:** 5

**Summary:**

This paper proposed a few-shot 3DGS model to achieve novel view synthesis with sparse view input. They improve the original 3DGS model mainly from the initialization perspect to mitigate the overfiting with sparse view input. Besides the initialization, they still designed some regularization strategies to further improve the model.

**Strengths:**

1. The overall writting is clear.
2. The notivation to improve the initialization of 3DGS in the sparse situation is reasonable.
3. The artical structure is easy to understand.

**Weaknesses:**

1. Improving the initialization of 3DGS in bothe dense and sparse situation is not the new ideal, e.g., InstantSplat [1] adopts the DUSt3R to get denser and more accurate points, SCGaussian [2] and FewViewGS[3] adopt the feature matching model to get more accurate initialization points.
2. The novelty of the declared "low-frequency awere SFM" is questionable. The matching method SFM is existing classic method, and this strategy mybe more like a superoir configuration of SFM tool.
3. The comparisons lack many of the latest methods, including [1,2,3]. And the results still don't show the obvious performance advantage.
4. The ablation results of each design do not show sufficient difference to prove their effectiveness.


[1] InstantSplat: Sparse-view Gaussian Splatting in Seconds.

[2] Structure Consistent Gaussian Splatting with Matching Prior for Few-shot Novel View Synthesis.

[3] FewViewGS: Gaussian Splatting with Few View Matching and Multi-stage Training.

**Questions:**

Many recent generation methods like CAT3D [1] and SEVA [2] or the large reconstruction models like lvsm [3] have shown greater potential in the sparse input, maybe this paper further needs to add some discussion about the comparison with these methods.

[1] CAT3D: Create Anything in 3D with Multi-View Diffusion Models.

[2] Stable Virtual Camera: Generative View Synthesis with Diffusion Models.

[3] LVSM: A Large View Synthesis Model with Minimal 3D Inductive Bias.

---

### Official Review · Reviewer_xWWP · 2025-11-01

**Soundness:** 2
**Presentation:** 2
**Contribution:** 2
**Rating:** 4
**Confidence:** 4

**Summary:**

The paper introduces a sparse-view 3D Gaussian Splatting (3DGS) method, which primarily focuses on the initialization point cloud. The authors propose 1. a frequency-aware SfM that augments the points in low-frequency areas. 2. Additional point cloud are extracted from a lightly trained 3DGS model.  3. point cloud regularizations that focuses on point cloud filtering, denoising, and normal consistency.

**Strengths:**

1. The paper emepirically find the bottleneck of 3DGS performce lies in the initialization.
2. The proposed method combined with DropGS outperform other methods in most metrics on Mip-NeRF360 and LLFF datasets.

**Weaknesses:**

1. The quantitative improvements brought by point cloud regularization are moderate based on Table 2. A qualitative figure would help understand the effectiveness of proposed method better.
2. The paper misses a reference SPARS3R [1], which also focuses on 3DGS initialization.

[1]Tang, Y., Guo, Y., Li, D., & Peng, C. (2025). SPARS3R: Semantic Prior Alignment and Regularization for Sparse 3D Reconstruction. In Proceedings of the Computer Vision and Pattern Recognition Conference (pp. 26810-26821).

**Questions:**

1. What are the limitations of the proposed method?
2. Some zoomed areas in Figure 3 seem at wrong positions. The significant digits are off for lpips metric in Table 2. The drafts merits a careful proofreading.

---

### Official Review · Reviewer_i3JL · 2025-11-11

**Soundness:** 3
**Presentation:** 3
**Contribution:** 3
**Rating:** 4
**Confidence:** 4

**Summary:**

This paper investigates the root causes of overfitting in sparse-view 3D Gaussian Splatting (3DGS) and concludes that initialization quality, rather than training-time regularization, is the key factor governing reconstruction quality. To address this limitation, they propose a three-stage initialization pipeline. First, it enhances Structure-from-Motion (SfM) with low-frequency-aware masking and relaxed track constraints to improve coverage in low-texture regions. Second, 3DGS Self-Initialization converts photometric cues into additional 3D points for denser geometry. Third, Point-Cloud Regularization applies single-view filtering, clustering-based denoising, and normal-consistency filtering to ensure geometric reliability. Experiments on Mip-NeRF360 and LLFF datasets demonstrate consistent improvements over prior baselines (FSGS, CoR-GS, DropGS, EAP-GS). The method achieves state-of-the-art reconstruction quality and delivers further performance gains when combined with existing regularization methods, all while maintaining comparable computational efficiency.

**Strengths:**

1. Rigorous ablation studies show that initialization, not regularization, drives performance in sparse-view 3DGS, an important empirical finding for the community.
2. Achieves state-of-the-art results across multiple benchmarks with superior quantitative and qualitative performance.
3. Integrates seamlessly with existing methods (e.g., DropGS) to further boost performance.

**Weaknesses:**

1. Lack of failure analysis: The paper does not discuss potential failure cases where self-initialization or clustering may introduce artifacts or erode fine structural details.
2. Limited novelty: The contributions primarily combine empirical observations with incremental engineering refinements to enhance SfM-based initialization for 3DGS, rather than introducing a fundamentally new algorithmic concept.
3. Unassessed SfM accuracy: Increasing SfM density by reducing the track length from three to two may introduce numerous false correspondences, yet the paper provides no quantitative evaluation of SfM accuracy. Excessive noise in initialization could potentially harm 3DGS reconstruction quality, but this trade-off is neither analyzed nor reported.
4. Scalability concerns: The method’s robustness under extremely sparse inputs (e.g., fewer than three views) or in large-scale outdoor scenes is not evaluated, leaving its scalability uncertain.
5. Incomplete evaluation: The presented qualitative results appear selective, lacking comprehensive multi-view or video-based comparisons to demonstrate consistent improvements beyond selected views or regions.

**Questions:**

Reducing the track length to 2 in Structure-from-Motion increases point density but compromises reliability. Two-view triangulation introduces weaker geometric constraints, making reconstructed points more prone to noise, depth ambiguity, and mismatched features. Without multi-view redundancy, outliers and depth errors become difficult to detect and correct, often resulting in a noisier and less consistent point cloud. It remains unclear how the authors ensure that their subsequent processing and denoising steps can effectively mitigate these issues to produce a clean and accurate initialization. I would love to hear more from the author about this.

---

### Note · Authors · 2025-11-22

I have read and agree with the venue's withdrawal policy on behalf of myself and my co-authors.